# Towards Concept-Aware Large Language Models

**Chen Shani**[ℵ]**, Jilles Vreeken**[⋆]**, Dafna Shahaf**[ℵ]

[ℵ] The Hebrew University of Jerusalem, Israel
[⋆] CISPA Helmholtz Center for Information Security, Germany

[ℵ] {chenxshani,dshahaf}@cs.huji.ac.il
[⋆] vreeken@cispa.de

## Abstract

Concepts play a pivotal role in various human cognitive functions, including learning, reasoning and communication. However, there is very little work on endowing machines with the ability to form and reason with concepts. In particular, state-of-the-art large language models (LLMs) work at the level of *tokens*, not concepts.

In this work, we analyze how well contemporary LLMs capture human concepts and their structure. We then discuss ways to develop concept-aware LLMs, taking place at different stages of the pipeline. We sketch a method for *pretraining* LLMs using concepts, and also explore the simpler approach that uses the output of existing LLMs. Despite its simplicity, our proof-of-concept is shown to better match human intuition, as well as improve the robustness of predictions. These preliminary results underscore the promise of concept-aware LLMs.

## 1 Introduction

Concepts are the glue that holds our mental model of the world together. It is hard to see how any intelligent agent could do without them. While there is no agreed-upon definition of concepts, one can think of them as the *mental representations* that enable us to identify objects and events as belonging to certain categories, communicate about them, and comprehend new situations in terms of previous ones: when we encounter a new situation (e.g., restaurant), we draw inferences about it using concepts we have already formed ("menu", "waiter").

Concepts can be concrete ("soup") or abstract ("tasty"). They can also be complex, e.g., "good winter beach destinations". While there is a lively debate on their exact nature, researchers agree **concepts play a pivotal role in various cognitive skills** such as reasoning, categorization, learning, planning, and decision-making (Murphy, 2004).

Thus, they are of interest to AI researchers wishing to endow machines with such abilities.

The representation of concepts has been studied in NLP, ML, and knowledge representation (Fumagalli and Ferrario, 2019; Davis and Marcus, 2015; Gardenfors, 2014; Speer et al., 2017), where they often view concepts as fixed, shallow structures representing some set of entities. For example, in the work on Chen et al. (2020) concepts are flat sets of context-independent entities. However, recent studies suggest concepts are more flexible and dynamic (Gabora et al., 2008); unfortunately, AI still struggles with accounting for the creative, context-sensitive manner in which people employ concepts.

In this work we focus on adding concepts to large language models (LLMs). Recently, LLMs (Yang et al., 2019; Raffel et al., 2020; Thoppilan et al., 2022; Scao et al., 2022; Zhang et al., 2023; Bubeck et al., 2023) gained immense popularity, achieving SOTA results across the board. However, they all work at the level of *tokens*, not concepts.

This is problematic for even the most fundamental LLM task – perplexity-based *text completion*. Ranking by string probability is distorted by *surface form competition*: different tokens compete with each other, even if they represent the same concept ("mother" and "mom") (Holtzman et al., 2021). In other words, the probability mass of a concept is distributed across many different tokens, distorting the ranking.

The problem runs deeper than mere synonyms. For example, consider the sentence "I can't get home for the holidays because of the [MASK]." The completions "snow", "blizzard", "weather", and "slippery roads" are not synonyms per se, but they correspond to the same concept – bad weather leading to hazardous driving conditions – and we believe that an LLM should treat them as such.

We stress that concepts are context-dependent; for example, while "snow" and "blizzard" are similar in the context of the sentence above, they

are very different for the sentence "I love eating [MASK] cones."[1] Thus, we cannot rely on knowledge bases (such WordNet (Miller, 1995)) for generating static, context-free concepts to be used for training LLMs.

We take the first step towards **concept-aware LLMs**, exploring the following questions:

**RQ1:** How well do LLMs capture concepts?

**RQ2:** How well do LLMs match human organization of concepts?

**RQ3:** How can we enhance an LLM in terms of concepts, with or without retraining?

We first show that contemporary LLMs capture human concepts to some extent, but they are still far from humans (RQ1). We then find that LLMs violate many of the principles of human concept organization, exhibiting some inconsistency (RQ2).

Lastly, we explore RQ3 from two different angles: first, we sketch a method to pretrain concept-aware LLMs. Next, we implement a proof-of-concept model-agnostic method to shift any off-the-shelf pretrained LLM from token- to concept-level with no further training. Our method improves both the ranking and robustness of the underlying LLM.

While we present here merely a promising proof-of-concept, our underlying objective of **endowing LLMs with concepts holds tremendous promise for the next-generation of LLMs**. We hope our work will spur further research, paving new roads into this exciting new territory.

## 2 RQ1: How well do LLMs capture concepts?

In this section, we explore to what extent LLMs grasp concepts. While concepts can be abstract and complex, here we focus on concrete and simple ones, since these are the basic building blocks of human concepts (Varela et al., 2017).

**Dataset.** To probe LLM abilities, we use the 100 everyday things (ETs) dataset (Gu et al., 2022), containing natural and man-made items everyone should be familiar with (egg, dog, elevator, table, etc.).

Perhaps the most basic relation between concepts is `TypeOf` (also called `IsA`). This relation governs the hierarchical organization of concepts humans possess. For example, we all know that a sandal is a type of shoe (denoted as `TypeOf`(sandal, shoe)).

To automatically extract `TypeOf`-relations we extracted the direct hypernyms and hyponyms of the 100 ETs' first sense using WordNet.[2] This results in data of the following form (ET in bold):

- {footwear} ← **shoe** ← {anklet, baby shoe, moccasin, oxford, sandal, running shoe, ...}
- {canine, domestic animal} ← **dog** ← {newfoundland, hunting dog, dalmatian, corgi, ...}

On average, each ET had 9.5 hyponyms and 1.1 hypernyms. Note that WordNet is noisy, and sometimes the first sense is not the intended one. We added a human baseline to show the data is of satisfactory quality.

**LLM probing.** We test the concept-`TypeOf` hierarchy of four representative LLMs: BERT-base-uncased, T5-large, GPT-davinci-003 (Legacy), and GPT-4.[3]

Since GPT-based models support question-answering, we use binary questions: "Is <ET> a type of <hypernym>?", and "Is <hyponym> a type of <ET>?"[4]

BERT and T5 are not optimized for question-answering; thus, we query them as follows: "<ET> is a type of [MASK].", and "<hyponym> is a type of [MASK].", and search for the desired hypernym and ET respectively along the top-k completions. The LLM is correct only if the answer is within the top-k completions.[5]

As a sanity check, we added a human baseline using a random sample of 100 `TypeOf`(ET, hypernym) and 100 `TypeOf`(hyponym, ET) questions used for querying the GPT models. To balance a bit the answer distribution, We added 20 negative examples per question type (of the form: "Is <ET> a type of <another ET's hypernym>?", "Is <hyponym> a type of <different ET>?"). The negative examples were not part of our analysis and were included to avoid annotators seeing only positive examples. We used six members of our research group for the annotation. Their mean response variance is 0.18 for the hyponyms and 0.07 for the hypernyms, showing they are calibrated.

---

[1] Snow cones are shaved-ice desserts. Blizzard cones are apparently beak-shaped face masks from the 1930s.

[2] https://wordnet.princeton.edu/

[3] We ran the GPT-4 experiments on October 2023.

[4] We queried both GPT models on 200 random combinations of "Is <ET1> a type of <ET2>?" ("Is dishwasher a type of tent?"). It returned "yes" 0(!) times.

[5] We are aware that this task is potentially harder than GPT's. However, testing GPT as if it were a masked LLM would likely lead to sub-optimal performance. We hence prefer to explore two different scenarios, increasing the robustness of the analysis, rather than artificially leveling the playing field.

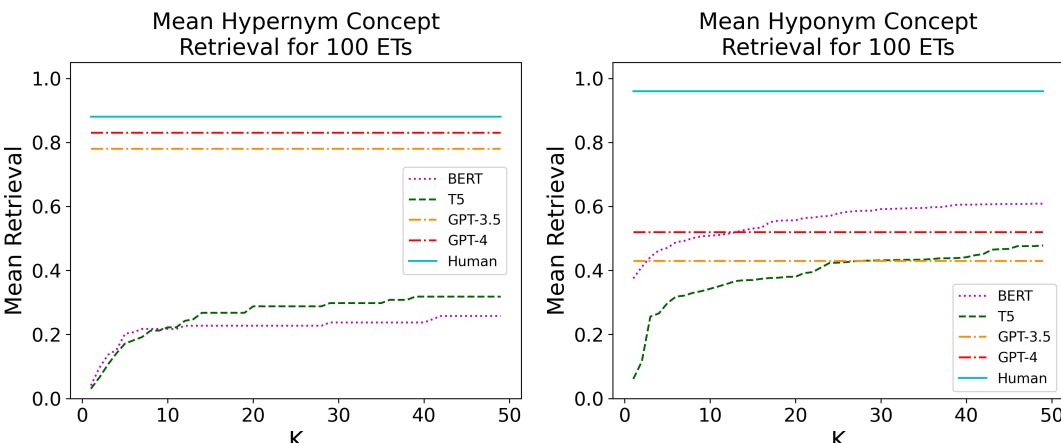

Figure 1: [Higher means better] LLMs concept retrieval as a function of K. For each ET, we measure how well it retrieves in its top-k completions the ET's hypernyms (left plot) and hyponyms (right plot). Since GPT and humans answer yes/no questions, their performance does not change as a function of K.

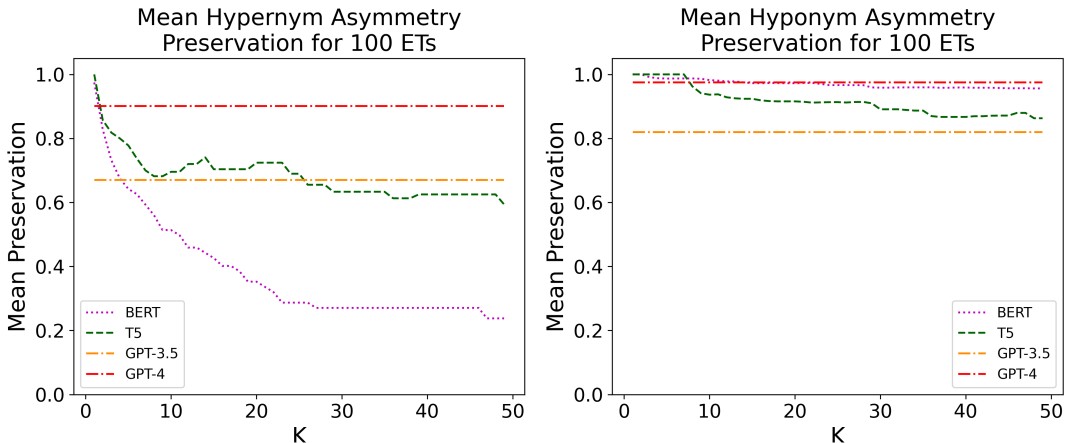

Figure 2: [Higher means better] LLMs' asymmetry preservation as a function of K. For each ET, we measure how well it preserves asymmetry by *not* returning the relevant item in its top-k completions (measured only using `TypeOf` relations the LLMs correctly retrieved in RQ1). We probe both for hypernyms (left plot) and for hyponyms (right plot). Since GPT supports question-answering, its performance does not change as a function of K.

**Results.** We plot the accuracy (or concept retrieval, as all examples are positive), as a function of $k$ for all four LLMs and humans (Figure 1). The results indicate that LLMs capture concepts to some extent, but it is far from perfect and human baseline. Models achieved (at k=50) 30%-83% retrieval for the hypernyms (human baseline is 88%) and 43%-60% for the hyponyms (human baseline is 96%). Interestingly, the GPT-based models seem to grasp more general concepts better (hypernym level), whereas BERT and T5 perform better on more specific concepts (hyponym level).

## 3 RQ2: How well do LLMs match human organization of concepts?

In this section, we explore concept *organization*. We focus on three agreed-upon organization princi-

ples of the human `TypeOf` hierarchy of concepts: *asymmetry*, *transitivity*, and *property inheritance*.

> **Asymmetry**
> $$\texttt{TypeOf(A, B)} \implies \neg \texttt{TypeOf(B, A)}$$

The human `TypeOf` relation between concepts is asymmetric; if sandals are shoes, shoes are not sandals. To measure how well an LLM preserves asymmetry we take all the `TypeOf` relations it identified in RQ1 and query the *other direction*. If an LLM correctly retrieved `TypeOf`(sandal, shoe), we now query for `TypeOf`(shoe, sandal) using the same querying methods and LLMs. If the LLM did not retrieve this other direction – it successfully preserved the asymmetry principle.

In Figure 2 we see that all four LLMs preserve

asymmetry better for hyponyms compared to hypernyms (stronger trend for BERT and T5). Overall, LLMs do preserve asymmetry, but they are still far from a complete understanding of directionality.

> ## Transitivity
> $$\text{TypeOf(A, B)} \land \text{TypeOf(B, C)}$$
> $$\implies \text{TypeOf(A, C)}$$

TypeOf is a transitive relation; if TypeOf(shoes, footwear) and TypeOf(sandals, shoes), then TypeOf(sandals, footwear). To measure whether LLMs preserve transitivity, we collect for each LLM all the pairs it correctly identified both {TypeOf(ET, hypernym), TypeOf(hyponym, ET)}. We then query the LLM for the relation between the hyponym and the hypernym directly (skipping the middle level of the ET). We use the queries "Is <hyponym> a type of <hypernym>?" (for GPT), and "<hyponym> is a type of [MASK]" (BERT and T5; we then search for the hypernym along the top-k=50 completions).

Figure 3 depicts the results. All models exhibit some level of transitivity, with GPT performing best, followed closely by BERT (starting around k=15). Interestingly, GPT-3.5 performs slightly better than GPT-4.

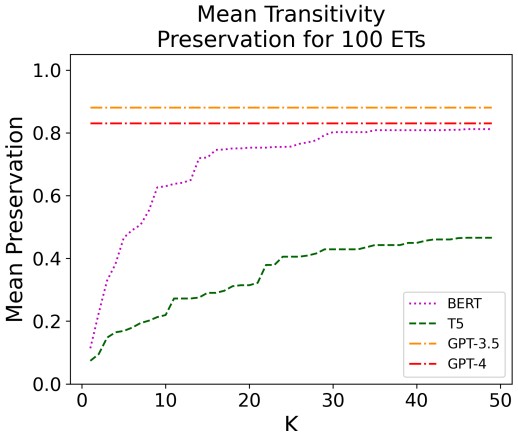

Figure 3: [Higher means better] Transitivity as a function of K for TypeOf relations correctly retrieved.

> ## Property inheritance
> $$\text{TypeOf(A,B)} \land \text{R(B)} \implies \text{R(A)}$$

If a property is true for a concept, it should hold for concepts below it in the TypeOf hierarchy. For example, if all footwear is manufactured, then all sandals are manufactured as well.[6]

---

[6]Note this has exceptions, e.g., penguins are birds and

| LLM | R(hyper) $\overset{?}{\to}$ R(ET) | R(hyper) $\overset{?}{\to}$ R(hypo) | R(hyper) $\overset{?}{\to}$ R(hypo) \| R(ET) |
|---|---|---|---|
| BERT | **0.85** | **0.80** | 0.92 |
| T5 | 0.56 | 0.68 | **0.93** |
| GPT-3.5 | 0.73 | 0.72 | 0.85 |
| GPT-4 | 0.73 | 0.73 | 0.85 |

Table 1: [Higher means better] Property inheritance: when the LLM knows that property R holds for a hypernym of an ET, we check how often it knows that R holds for the ET (left), for its hyponyms (middle) and for its hyponyms given that it knows R(ET) holds (right).

To test for LLMs' property inheritance, we first enrich our dataset with attributes of the hypernyms. We used Quasimodo, a commonsense knowledge base that focuses on salient properties that are typically associated with certain objects or concepts (Romero et al., 2019). It consists of (object, predicate, subject) triplets, e.g., "(footwear, affect, skeletal system)", "(footwear, has property, manufactured)". To ensure high-quality properties, we set the triplet saliency score threshold to 0.9. We collected 771 triplets for 37 ETs. The mean number of triplets per ET is 20.84 (STD=26.14).

We use the first paraphrase of wordtune[7] to turn triplets into sentences and mask the object: "(footwear, affect, skeletal system)" $\to$ "The skeletal system is affected by footwear." $\to$ "The [MASK] is affected by footwear.".

For each sentence, we query the LLMs to see if they believe it to be true (k=50 for BERT and T5). For each LLM, if it believes a hypernym triplet holds, we also query it by replacing the hypernym with the ET (Table 1 left column) and with its hyponyms (middle column).

We modified the sentences to questions for GPT by adding "?" at the end of the paraphrased sentence, and taking the first proposed paraphrase ("Does footwear affect the skeletal system?").

Overall, all models show some level of property inheritance (Table 1). Interestingly, all models are more likely to transfer a relation from the hypernym to its hyponym if they correctly transferred it from the hypernym to its ET (middle vs. right columns), showing some robustness in LLMs' understanding of transitivity. Also worth noting – BERT seems to preserve property inheritance better than the rest of

---

birds can fly, but we have yet to discover a flying penguin.
[7]https://www.wordtune.com/

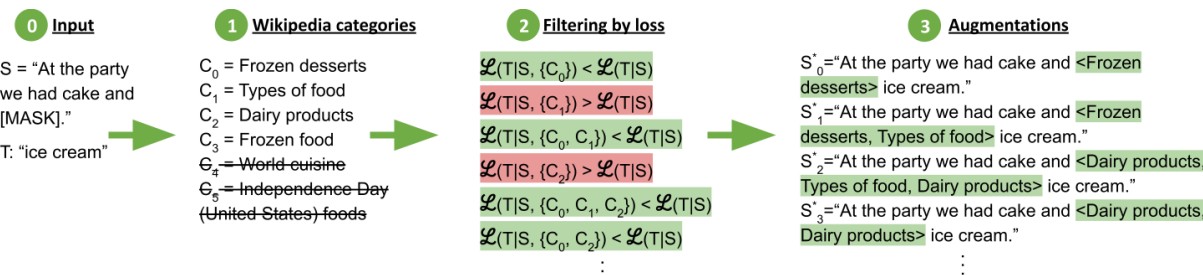

Figure 4: Overview of our sketch for a method to pretrain concept-aware LLMs. The model is trained to choose the categories that will best assist it in predicting the missing token(s).

the models.

To conclude RQ2, GPT models seem consistently better than BERT/T5 for asymmetry and transitivity (for inheritance property BERT takes the lead), but all models do violate principles of human organization and exhibit some inconsistencies.

## 4 RQ3: How can we enhance an LLM in terms of concepts, with or without retraining?

In the previous sections, we have shown that LLMs' understanding of concepts is still limited. As we noted earlier, concepts play a pivotal role in various human cognitive skills, including reasoning, categorization, planning, and decision-making; we strongly believe that equipping LLMs with a notion of concepts can significantly expand their capabilities, bringing them closer to human performance.

Endowing LLMs with concepts can take many forms, and may occur at different stages (pretraining, fine-tuning, post-processing). We now lay the ground for future studies on concept-aware LLMs. We start by proposing a sketch of a method for *training* concept-aware LLMs. We then take the simpler, proof-of-concept approach of building concepts on top of the output of existing LLMs.

**Approach 1: Rethinking LLM training**

We envision training a model to **choose a subset from a closed set of concepts** that will assist it the most in completing a missing sequence of text. Potential concepts could come from Wikipedia categories, or perhaps some knowledge graph.

See our suggested approach in Figure 4. Given an input sentence, we sample a span of text $T$ that can be linked to a Wikipedia article (similar to Wu et al. (2019)), such as "ice cream", and mask it. We denote by $S$ the masked sentence. We compile a subset of Wikipedia categories that the missing text $T$ belongs to and their parent categories, filtering

out categories that are too narrow ("Independence Day (US) foods") or too wide ("World cuisine"), resulting in a set of candidate concepts $C$.

Next, we iterate over all possible subset combinations of $C$ (or perhaps sample from them, if the set is too big) and interleave them with the text using a special token, resulting in augmented sentences $\{S'_i\}$ such as "At the party we had cake and <Dairy food, Frozen desserts> ice cream". We then mask $T$ in all augmented sentences $\{S'_i\}$ and filter out sentences that do not reduce the prediction loss over $T$, compared to the non-augmented masked sentence $S$. That is, we only keep the original (non-masked) sentences augmented with subsets of concepts that help predict the missing text $T$.

We suggest to train an LLM over the augmented, non-masked sentences $\{S'_i\}$ corresponding to concepts that do reduce the loss, resulting in a **model that teaches itself when and which concept(s) to predict**, and how to **best incorporate the predicted concept(s)** into future token prediction.

The above suggestion is merely a sketch of the approach, and there are many possible variants. We can treat the problem of concept prediction as a probabilistic multi-class classification (over a closed set of categories) or as free text generation; we can even take an approach similar to Toolformer (Schick et al., 2023) and use external tools.

The sketch is inspired by the recent success of approaches that augment LLMs with additional information during pretraining (Aghajanyan et al., 2021; Schick et al., 2022; Izacard et al., 2022), and in particular by Toolformer (Schick et al., 2023), that adds additional information only if it is helpful.

**Approach 2: Building on top of existing LLMs**

We consider Approach 1 to be promising, but implementing it requires significant resources. Rather than rethinking LLM architecture and training, we now consider the other end of the spectrum, build-

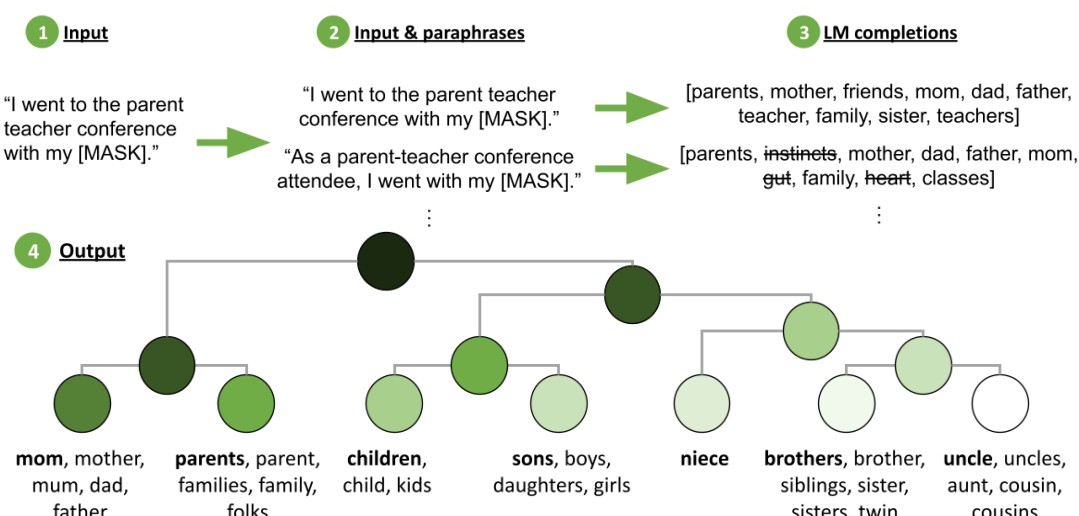

Figure 5: Overview of our algorithm for extracting concepts from pretrained LLMs. We augment the input sentence by paraphrasing and predict the top $k$ completions for each paraphrase. Next, we filter out rare and unlikely tokens (strikethrough) and perform agglomerative clustering using the LLM's token-contextual embeddings (centroid in bold). We assign new weights to each node in the dendrogram (darker ranked higher, sorted according to weight).

ing a concept-aware LLM on top of existing LLMs, without any further training or external data.

Previous works showed improved results of pretrained LLMs without further training on tasks such as word sense disambiguation, factualness and consistency (Levine et al., 2020; Liu et al., 2022). We believe our post-hoc method could similarly enhance downstream tasks.

**Concept-completion task.** We start by testing our ideas on the fundamental LLM task of text completion (fill-mask). Given a masked sentence $\mathcal{S}$ and an LLM, our goal is to return a ranked list of *concepts* $C_1, ..., C_N$. Each concept $C_i$ is a non-empty set of tokens $T$. Ideally, concepts and their ranking should correspond to human intuition.

Figure 5 illustrates our main idea. In short, given a masked sentence $\mathcal{S}_0$ ("I went to the parent teacher conference with my [MASK]."), we retrieve the LLM top completions, paraphrasing $\mathcal{S}_0$ as an augmentation technique to increase robustness. To form concepts, we perform agglomerative clustering using the LLM contextual embeddings.

For clarity of presentation, Figure 5 shows only a segment of the dendrogram, rather than going all the way to singletons. Nodes in the dendrogram have scores based on their tokens' weights and frequency in paraphrases (darker means higher). The bottom layer is sorted according to score. The first (bold) token in each node is the cluster centroid.

In this example, the most likely concept (left) contains tokens such as "mom", "mother" and "dad", followed closely by a concept containing "parents" and "family". Next concepts refer to children and other family members. As we go higher, concepts become more general; the top node in the figure roughly corresponds to "family member".

## 4.1 Algorithm

Figure 5 depicts our implementation.[8] We give a succinct overview, for details see the Appendix.

**Augmentation.** To augment the input sentence $\mathcal{S}_0$ we first retrieve the LLM top-k completions.[9] We replace the "[MASK]" token with the first completion that is not a stopword or a sub-word and paraphrase using wordtune.[7] We then re-mask the input sentence $\mathcal{S}_0$ and mask its paraphrases $\{\mathcal{S}_1, ..., \mathcal{S}_{M-1}\}$, resulting in $M$ masked sentences.

**Top-k completions retrieval.** We retrieve the top-k (k=100) completions for each sentence in $\{\mathcal{S}_0, ..., \mathcal{S}_{M-1}\}$. We count how often completions appear and remove infrequent ones.[9] We extract the contextual embeddings (the token embedding from the last hidden layer using the masked input sentence $\mathcal{S}_0$ with the corresponding token as completion). We use the contextual embedding, as different tokens may belong to the same concept or not, depending on the context. The fact that the LLM's embedding yields meaningful clusters hints it somewhat captures concepts, which is in

---

[8] Our code can be found at https://github.com/chenxshani/Towards-Concept-Aware-LLMs

[9] See details in the Appendix.

line with our findings from RQ1.

**Clustering & Ranking.** We reduce the dimensionality using PCA and t-SNE,[9] and use agglomerative clustering to cluster the completions into concepts. We use agglomerative clustering as different thresholds yield different concept-granularity, similar to the flexibility of concepts in humans. Each cluster is assigned with a weight that corresponds to: 1) the token with the maximal soft-max score, to avoid problems related to surface form competition, and 2) the token with the maximal number of repetitions across augmentations' top-100 completions, to increase robustness (a token that repeated frequently is probably very relevant).[9]

## 4.2 Evaluation

To evaluate our method, we focus on *fill-mask* task (completing a masked sequence of text).

**Experiments.** We use the ProtoQA dataset, consisting of questions regarding prototypical situations (e.g., "Name something you are likely making if you buy milk, eggs, sugar and cream.") (Boratko et al., 2020). We believe this setting is relevant for our use case, as there are usually multiple relevant answers. To make the input similar to the language LLMs are usually trained on, we manually changed the questions to first-person statements ("I bought milk, eggs, sugar and cream to make a [MASK]."). We used 63 sentences to set our hyper-parameters and an additional 100 sentences for evaluation.[9]

We used BERT-base-uncased, the most popular fill-mask model (Devlin et al., 2018).[10] We compute both BERT's top 100 completions and our manipulation, which we call concept-BERT, on top of BERT's output, resulting in ranked *concepts*.

### 4.2.1 Cluster quality

We measure the semantic coherence of clusters using the cosine similarity of word2vec's token embedding (first ten clusters for all sentences). The mean **within-cluster** similarity is 0.41, whereas the mean **inter-cluster** similarity is 0.12. For reference, BERT's top-ten completion similarity is 0.22. Hence, our clusters are coherent and distinct.

A closer examination highlights the distinction between the next-token-prediction approach and ours. Consider the sentence "I can't get home for the holidays because of the [MASK]." and its cluster: {blizzard, cold, temperature, snowfall,

---

[10]Most common according to Hugging Face: https://huggingface.co/models?pipeline_tag=fill-mask.

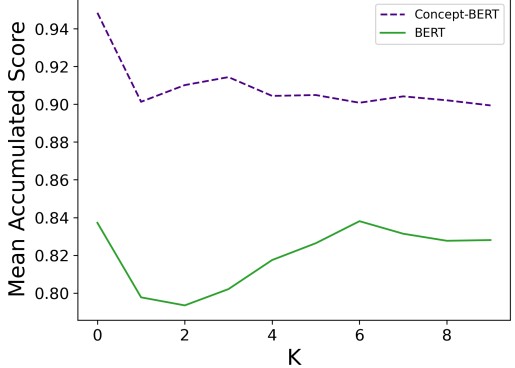

Figure 6: [Higher means better] Concept-BERT's and BERT's mean score at k=10. K denotes the number of clusters for Concept-BERT and the number of tokens for BERT. Our method's mean score at k=1 is 95% whereas BERT's is 84%. Concept-BERT is consistently better than BERT.

weather, snow}. While the concept is coherent (i.e., cold weather conditions), some *tokens* are less-natural completions without their cluster-context (e.g., "temperature").

### 4.2.2 Ranking quality

We evaluate the quality of ranking by annotating all completions in the top-10 concept-BERT *clusters* and top-10 BERT tokens for all 100 input sentences. Three crowdworkers received the masked sentence and a possible completion, and were asked to classify the completion as either: likely (score=1)/ possible but unlikely (score=0.5)/does not make sense (score=0). See qualifications, compensation, and instructions in the Appendix. Note this evaluation cannot be automated, as we wish to see if our concept-aware modification aligns the LLM's output with *humans*. A completion's aggregated score is its mean score across the three annotators (mean-variance across annotations=0.17). For concepts, we average all tokens in the cluster. Our **score at k=1 is 95%** whereas BERT's is 84% (see Figure 6). Moreover, for all k values, **concept-BERT is consistently better than BERT**.

### 4.2.3 Completions in dispute

We now focus on completions for which BERT and concept-BERT **disagree** – one predicts the completion is likely, while the other does not (and vice versa). We believe these are the most interesting regions to evaluate our manipulation on. See examples of disputed completions in Table 2.

We treat the middle 15% of the ranked lists as

| Input sentence | Completion | BERT | Concept-BERT |
|---|---|---|---|
| I bought a fake [MASK] from a street vendor. | jersey | 0.08 | 0.79 |
| When I retired I started [MASK]. | cycling | 0.06 | 0.77 |
| Whenever I suffer from cold I always [MASK]. | shudder | 0.04 | 1 |
| | rise | 0.93 | 0.24 |
| When I go to the beach I use [MASK] to protect myself from the sun. | sticks | 0.91 | 0.28 |
| | soap | 0.74 | 0.03 |
| I always take my [MASK] with me to the gym. | laptop | 0.76 | 0.29 |
| I squeezed myself into the [MASK]. | sand | 0.71 | 0.23 |

Table 2: Examples of completions for which the weight BERT and concept-BERT assign are notably different. Our manipulation increases the relative rank of appropriate completions and decreases the rank of inappropriate ones. Relative rank calculation: $(1 - \text{completion rank})/K$ where K=100 for BERT and k=number of outputted clusters for concept-BERT. Color coding: red=low score, orange=intermediate, green=high.

| Scenario | Mean score | Norm. score |
|---|---|---|
| Concept-BERT ↑ BERT ↓ | **0.84** | **0.304** |
| Buffer | 0.74 | - |
| Concept-BERT ↓ BERT ↑ | 0.66 | -0.142 |

Table 3: [Higher means better] Mean scores and normalized (using the buffer) scores of the three scenarios in the dispute evaluation. Tokens concept-BERT ranked as probable while BERT as improbable (first row) are annotated significantly better than both the buffer (middle row) and the tokens BERT ranked high and concept-BERT low (bottom).

buffer and output tokens that are above the buffer according to one model and below according to the other. This way, we identified 282 disputed tokens. In addition, we annotated completions that both models ranked in the middle 15% (buffer). Volunteer computer science graduate students annotated 585 completions using the same setup as in §4.2.2 (282 disputed completions and 303 buffer completions). Each completion was annotated by two students (mean-variance across annotations=0.1).[9]

We divide the annotated completions into three groups and compute their mean annotation scores. As some sentences have more good completions than others, we also normalize the mean score per sentence by subtracting the sentence buffer's mean score. Table 3 shows that **when the models disagree, concept-BERT is more often correct** (also refer to Figure A.7 in the Appendix)

Next, we compute the accumulated mean accuracy of completions as a function of rank given by each of the models (Figure A.8). We expect a negative correlation since the quality should decrease when going down the ranked list. While concept-BERT does have a negative correlation, BERT is actually *positively correlated* (meaning, its top-ranked completions are on average worse than the bottom-ranked ones). Both curves have significant correlation (p-values<0.05), whereas BERT's is weaker (coefficient $0.54$ versus $0.91$). We stress this is not a random sample, but rather the disputed completions (and buffer). Thus, our concept-aware manipulation reveals appropriate completions and removes inappropriate ones, with respect to the original LLM.

Lastly, we also analyze the mean accuracy of the disputed completions as a function of how strict the threshold for "in dispute" is. BERT's accuracy decreases much more sharply compared to concept-BERT ($> 10\%$ versus $< 2\%$), hinting our manipulation increases robustness (Figure A.9).

To conclude, our simple implementation led to overall coherent and distinct concept-clusters with meaningful ranking that improve the original ranking in a robust manner. We see promising indications that similar techniques could enhance LLM robustness.

## 5 Related Work

There is little work on endowing LLMs with a notion of concepts. The work that is closest to ours in spirit is SenseBERT (Levine et al., 2020), which shifts BERT from the form- to the *sense-level* by injecting WordNet token information, improving word sense disambiguation. Unlike SenseBERT, we suggest learning the categories in a *context-dependent manner* (by using the categories that best assist in predicting the missing text), rather than their static, context-independent method.

Works on knowledge base embeddings learn representations from knowledge graphs (Bordes et al., 2013), but they are also limited to simple concepts and KB relations, with no external context.

Aharoni and Goldberg (2020) showed that LLMs capture the domain a sentence belongs to. While domains can also be thought of as concepts, we are after a different (and dynamic) granularity.

ConceptX is an interpretability framework to analyze how latent concepts are encoded in representations learned within pretrained LLMs (Sajjad et al., 2022). They found that roughly 50% of the encoded concepts adhere to their suite of human-defined linguistic concepts, which aligns with our results from RQ1. However, their method fails to address complex, multi-faceted concepts.

Hanna and Mareček (2021) specifically explored the extent to which BERT captures hypernymy, exploring different prompts and evaluation metrics. We note that their dataset was somewhat biased towards Australian and New Zealand culture, and also contained multiple errors.

# 6 Conclusions & Future Work

In this work we explored the possibility of concept-aware LLMs, inspired by the importance of *concepts* in human cognition. Today's LLMs all work at the level of *tokens*, not concepts; this is problematic, as different tokens from the same underlying concept compete for the probability mass. Instead, we envision LLMs that first zero in on a likely concept, and only then pick the best surface form among the candidates.

We started by analyzing to what extent contemporary pretrained LLMs capture concepts and match the human organization of concepts. We showed that LLMs do capture concepts to some (far-from-perfect) extent, while sometimes violating the human concept-organization principles of asymmetry, transitivity, and property inheritance.

Next, we explored directions for augmenting LLMs with concepts from two different angles. We first sketched our vision for *training concept-aware LLMs*. Next, we presented a *model-agnostic* method to shift any off-the-shelf pretrained LLM from the token- to the concept-level, without fine-tuning or adding any external information. We showed that our concept-BERT (built on top of BERT) outputs a ranked list of *concepts* which are relatively coherent and distinct, better match human intuition, and are more robust compared to

the underlying model.

The notion of concepts has been extensively studied in psychology, and we believe many of the questions posed there could inspire directions for future work. For example, Prototype theory states there is a graded degree of belonging to a concept (Rosch, 1975), which we have not taken into account here. Other works tackled the problem of complex concepts and the *composition* of concepts. For example, Rosch (1975) showed that concepts can be combined ("tall" + "man") in a context-dependent way (e.g., the height for a man to be considered tall is not the same as for a building). For this direction, commonsense sources such as the distribution-over-quantities dataset (Elazar et al., 2019) might prove helpful.

While this is only preliminary work, we believe that concept-aware LLMs hold immense promise for the next-generation of LLMs, and could benefit many downstream tasks around learning, planning, and reasoning. One place where we believe this approach will be particularly useful is whenever there is disambiguity (speech recognition, word sense, spell check, etc.). Consider rare words that need to be identified by automatic speech recognition: by working at concept-level, these words could be clustered together with completions that are likely given the rest of the sentence but do not sound similar to the audio, thus increasing the acoustic model's certainty score. We hope to draw the community's attention to this exciting new research area of concept-aware LLMs.

# 7 Limitations

In RQ1 & RQ2, for BERT and T5 we rely on exact string matching. This entails that if the LLM retrieved a synonym of the right answer, we do not detect it as a successful retrieval.

Another limitation is the usage of noisy automation. We rely on WordNet's first word-sense for the 100 ETs, which might not be the original intent. Quasimodo might also introduce wrong relations (even when using a strict threshold, as we did). Moreover, Wordtune's paraphrasing is another possible source of noise. We note that throughout the process we consistently sampled to verify the quality of the automatic parts, and found them satisfactory.

As for RQ3, our method heavily relies on the input LLM, and thus might preserve some of the LLM's biases. One might try to overcome these

biases, e.g., by injecting external knowledge.

Another limitation of our method is the usage of not just the LLM completions, but also their embeddings. This does not let us apply our method to LLMs that expose only their completion output (e.g., accessible via API).

Lastly, as we run the LLM several times and postprocess (paraphrasing extraction, dimensionality reduction, clustering, etc.), computation of our proof-of-concept model is somewhat slower than that of the underlying LLM. We note, however, that many of those operations are easily parallelizable.

## Acknowledgements

We thank the reviewers for their insightful comments. This work was supported by the European Research Council (ERC) under the European Union's Horizon 2020 research and innovation programme (grant no. 852686, SIAM).

In memory of the more than one thousand victims of the horrific massacre carried out by Hamas terrorists on October 7th, 2023.

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

| LLM | Mean retrieval | # ETs | # Relations |
|---|---|---|---|
| BERT | 0.79 | 15 | 169 |
| T5 | 0.47 | 18 | 141 |
| GPT-3.5 | 0.88 | 57 | 493 |
| GPT-4 | 0.83 | 56 | 562 |

Table 4: Mean retrieval of the transitivity property for the four LLMs. For each pair where the LLM correctly retrieved that "<ET> `TypeOf`<hypernym>" and "<hyponym> `TypeOf`<ET>", we queried to see if it will also retrieve that "<hyponyms> `TypeOf`<hypernym>". Higher means better transitivity abilities.

Francisco J Varela, Evan Thompson, Eleanor Rosch, and Jon Kabat-Zinn. 2017. The embodied mind. *(No Title)*.

Ledell Wu, Fabio Petroni, Martin Josifoski, Sebastian Riedel, and Luke Zettlemoyer. 2019. Scalable zero-shot entity linking with dense entity retrieval. *arXiv preprint arXiv:1911.03814*.

Zhilin Yang, Zihang Dai, Yiming Yang, Jaime Carbonell, Russ R Salakhutdinov, and Quoc V Le. 2019. Xlnet: Generalized autoregressive pretraining for language understanding. *Advances in neural information processing systems*, 32.

Chaoning Zhang, Chenshuang Zhang, Sheng Zheng, Yu Qiao, Chenghao Li, Mengchun Zhang, Sumit Kumar Dam, Chu Myaet Thwal, Ye Lin Tun, Le Luang Huy, et al. 2023. A complete survey on generative ai (aigc): Is chatgpt from gpt-4 to gpt-5 all you need? *arXiv preprint arXiv:2303.11717*.

**RQ2: Transitivity**

Table 4 depicts the mean retrieval of the transitivity property for the three LLMs.

**RQ2: Property inheritance**

BERT's coverage is rather low, leaving us with 29 ETs and 709 triplets to consider. T5 covered 52 ETs with 262 triplets. Both GPT models' coverage is 51 ETs and 748 triplets.

**RQ3: implementation details**

**LM.** We used BERT-base-uncased with the default parameters. We performed no training.

**Augmentation.** For this phase we first replaced the missing token with the LLM's most probable completion that contains more than three letters and is not a stop-word (using the *stopwords* list from *nltk.corpus* package). We inserted $\mathcal{S}_0$ to AI21's Wordtune paraphrasing model using the default parameters:

```
requests.post(
"https://api.ai21.com/studio/
v1/experimental/rewrite",
headers={"Authentication":
<ai21-private-token>}
json={"text": S_0,
"intent": "general"})
```

And extracted the text suggestions from the output JSON file. We then searched for the original completion and masked all sentences. Sentences in which we were unable to automatically find the word were dropped.

**Top-k completions retrieval.** We used $k = 100$ for each masked sentence. We drop each completion that did not appear in at least half of our augmentations. Note: another possible implementation would be a function of *unique* completions.

**Clustering & Ranking.** As a latent space representation of contextual token, we extract the LLM's token embedding for this token from the last hidden layer with the input sentence $\mathcal{S}_0$. We reduce the dimensionality of the embeddings from 768 to 100 using PCA (scikit learn implementation, n_components=100, svd_solver='full') and from 100 to 10 using t-SNE (scikit learn implementation, n_components=10, init='pca', perplexity=10, method='exact').

We cluster the embeddings after the dimensionality reduction using agglomerative clustering using the distance metric cosine similarity, linkage='linkage', distance threshold=0.45, n_clusters=None, and compute_distances=True (scikit learn implementation)

To rank the clusters, we used the following formula:

$$weight(C_i) = \alpha \cdot max_{weight}(weight(t) \; \forall t \in C_i)$$
$$+(1 - \alpha) \cdot max_{rep}(rep(t) \; \forall t \in C_i)$$

where $\alpha = 0.7$.

### RQ3: Human annotators

For both annotation tasks (computer science graduate students and Amazon Mechanical Turk), annotators were presented with a sentence and a possible completion and were asked "Do you think this completion makes sense?". Possible responses are: {likely, possible but unlikely, does not make sense}.

**Precision at k.** We used Amazon Mechanical Turk with the following qualifications: {HIT Approval Rate > 98, Number of HITs Approved > 5000,

Location is one of CA, GB, US (for English speakers)}. We also used a custom qualification using five example sentences and completions. Annotators were allowed to make one error in order to qualify. We paid annotators $0.02 per completion. Overall, we had 39 unique annotators.

Full instructions:

You will be presented with a sentence containing a missing word and a candidate word to fill-in the blank. Your role is to determine for each completion whether it is likely / possible but not likely / does not make sense at all. Note! If a completion is not grammatically correct ("I enjoy *raining*" instead of *rain*) that is fine, we do not care about grammar here. But if the sentence + completion is not a full sentence ("I enjoy *doing*") that is NOT fine, as the sentence is meaningless.
Example 1
Sentence: I went to the parent teacher conference with my _____.
Completion: parent
Desired response: Likely
Example 2
Sentence: I went to the parent teacher conference with my _____.
Completion: schedule
Desired response: Does not make sense
Example 3
Sentence: I went to the parent teacher conference with my _____.
Completion: grandfather
Desired response: Possible but unlikely
Explanation: While this is not the common scenario, it is still possible.
Example 4
Sentence: I went to the parent teacher conference with my _____.
Completion: mothers
Desired response: Likely OR Possible but unlikely

**Completions in dispute.** We recruited 8 volunteers, all are graduate students from the computer science department (same instructions as the Amazon Mechanical Turk experiment, see instructions above). Each student annotated about 150 completions (cutoff at the end of the sentence). Each completion was annotated by two students. Mean-variance across annotation=0.1, showing a fairly good quality of annotations (possible responses=$\{0, 0.5, 1\}$). Students reported the task to take about 15 minutes to complete.

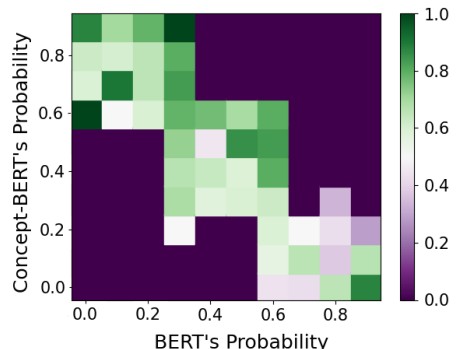

Figure 7: Heat-map of the disputed completions (higher means better). The y-axis represents concept-BERT's completion relative rank. The x-axis represents BERT's relative rank. The top-left part of the map received higher scores compared to the middle (buffer) and the bottom-right part. Meaning, our manipulation ranked high appropriate completions and low inappropriate ones.

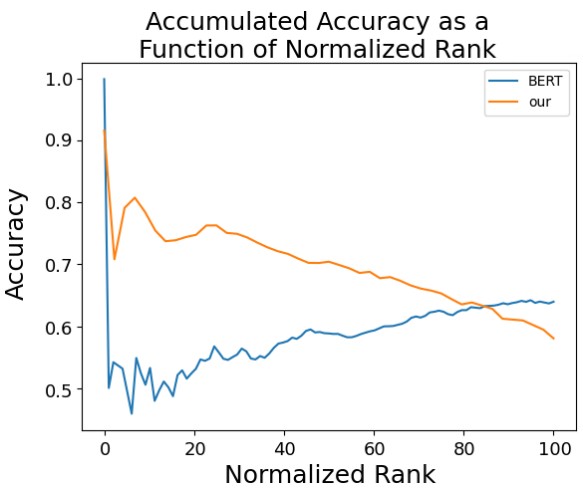

Figure 8: Completions' accumulated mean accuracy as a function of rank given by each of the models. Both curves have significant correlation (p-value $< 0.05$), whereas BERT's correlation is weaker (correlation coefficient $0.54$ versus $0.91$). Interestingly, while concept-BERT has a negative correlation, as expected since the rank's quality should decrease, BERT's correlation is positive. We stress this is not a random sample, but rather the disputed completions (and the buffer). Thus, this again strengthens our claim, that our manipulation helps to reveal appropriate completions and remove inappropriate ones, with respect to the original LLM.

**RQ3: Figures**

Figure 9: Mean accuracy of the disputed completions for both models as a function of how strict the threshold for disagreement is. BERT's accuracy decreases sharply compared to concept-BERT, suggesting our manipulation increases robustness.