# OpenReview forum: "Towards Concept-Aware Large Language Models"
_EMNLP/2023/Conference — EMNLP 2023 Findings_

### Official Review · Reviewer_YmXf · 2023-08-01

**Soundness:** 4

**Excitement:**

4: Strong: This paper deepens the understanding of some phenomenon or lowers the barriers to an existing research direction.

**Missing References:**

1. [Analyzing BERT’s Knowledge of Hypernymy via Prompting](https://aclanthology.org/2021.blackboxnlp-1.20) (Hanna & Mareček, BlackboxNLP 2021)

2. [Concept Understanding in Large Language Models: An Empirical Study](https://openreview.net/forum?id=losgEaOWIL7) (Liao et al., ICLR 2023)

3. [Better Language Model with Hypernym Class Prediction](https://aclanthology.org/2022.acl-long.96) (Bai et al., ACL 2022)

**Paper Topic And Main Contributions:**

Whether LLMs process concepts the same ways humans do is an underexplored question. Current state-of-the-art LLMs are trained to predict the next or masked _token_, which doesn't obviously induce concept-awareness-- probability mass of a concept will be spread over many tokens. The authors explore concept-awareness in pre-trained LLMs by (1) showing they capture concepts to some extent, but (2) often violate human organization of concepts such as transitivity. Then, the authors propose two mechanisms, one a pre-training sketch akin to Toolformer (Schick et al. 2023), and one off-the-shelf, where (3) the off-the-shelf procedure improves the concept-awareness of the LLM.

**Questions For The Authors:**

A. RQ2: Did you collect human judgments for asymmetry, transitivity, and property inheritance? Even though asymmetry should theoretically give 1.0, for example, perhaps for completeness the authors should include the corresponding human baseline (this comment did not affect my score)

B. l385 Which parameters are being finetuned? In l304 it is written "without any further training".

**Reasons To Accept:**

1. The research question (whether LLMs process concepts like humans) is timely, and I believe, important. The behavioral experiments, which are similar to those in Hanna and Mareček (2021), are straightforward and well-designed. The results make concrete the intuition that "LLMs capture concepts, but not well enough".
2. The paper is clear, enjoyable to read, and well-written (with some exceptions, see Weaknesses). I especially liked the broader framing on concepts and human cognition in the Introduction and Conclusions.

**Reasons To Reject:**

I have three reservations about the paper, one minor related to prior work (1), and then two regarding the engineering contribution. In particular, notes 2-3 would make the difference between a 3 and a 4/5 for Soundness.

1. Missing reference which uses almost the same prompting method to analyze hypernymy in BERT: [Analyzing BERT’s Knowledge of Hypernymy via Prompting](https://aclanthology.org/2021.blackboxnlp-1.20) (Hanna & Mareček, BlackboxNLP 2021)
2. l319 In practice, one directly uses LLM word embeddings or finetunes them on some downstream task. As the conceptBERT uses a frozen BERT model, however, the practical downstream application is unclear. In particular, the paraphrase procedure seems quite inefficient if the downstream application is similar to the one used for evaluation (ProtoQA), as the inference latency is higher for conceptBERT. To what extent is the inference latency unwieldy? These factors should be discussed in the paper.
3. Section 4.3: For completeness, experiments on GPT and T5 should also be included (perhaps in an Appendix).

**Reproducibility:**

4: Could mostly reproduce the results, but there may be some variation because of sample variance or minor variations in their interpretation of the protocol or method.

**Reviewer Confidence:**

4: Quite sure. I tried to check the important points carefully. It's unlikely, though conceivable, that I missed something that should affect my ratings.

**Typos Grammar Style And Presentation Improvements:**

### Major
I think the paper is well-organized and readable as-is. An alternative (this did not impact my score) could be to reduce and move the pre-training engineering sketch to later in the paper as it is more speculative, perhaps in Future Work or Discussion. Instead, this could be replaced with extended experiments on GPT and T5.

### Low-hanging fruit
1. Figure titles are often cut off at the top.
2. l145 Should $K$ be $k$?
3. l130 Should add: it is correct only if the answer is within the top-k completions. (Up to this point it is unclear)

---

> ### Author Rebuttal · Authors · 2023-08-29
>
> Thank you for your valuable insights. We are glad you found our research question timely, important, and well-framed, our experiments well-designed, and our paper clear and enjoyable to read.
>
> We thank you for pointing out the missing reference, and we will add it to the final version. We will also fix all the style issues.
>
> * “experiments on GPT and T5 should also be included (perhaps in an Appendix).”
>
> Our method requires a model that returns perplexities and conditional embeddings for the completions. Recent GPT implementations only offer partial access to this information. We acknowledge this limitation in the manuscript. We note it is possible to extend our framework to black-box models (e.g., by sampling multiple completions and rethinking the aggregation function), but we leave this for future work.
>
> We have not included T5 because it is an autoregressive model (the masked text has to be at the end of the input), but we have in fact experimented with T5-3b (only on sentences obeying the input requirement), and we have preliminary (promising) results that we will be happy to add to the Appendix.
>
> * Line 319 (“In practice, one directly uses LLM word embeddings or finetunes them on some downstream task. As the conceptBERT uses a frozen BERT model, however, the practical downstream application is unclear.”)
>
> We envision two main ways for incorporating concepts in downstream tasks.
> (1) We could fine-tune BERT on the downstream task and then build conceptBERT on top of the fine-tuned model (Approach 2). We posit the fine-tuning process will allow our manipulation to better identify concepts relevant for the downstream task.
> (2) Alternatively, Approach 1 suggests incorporating concepts into the training process itself; while we had pre-training in mind, this approach is also applicable to fine-tuning.
>
>
> * “[T]he inference latency is higher for conceptBERT. To what extent is the inference latency unwieldy?”
>
> Roughly speaking, this is our inference process for conceptBERT:
> 1) BERT prediction (~latency of the original inference)
> 2) Paraphrase generation (we use word-tune, but can be implemented locally to reduce the latency to milliseconds)
> 3) BERT prediction on N sentences (embarrassingly parallel; ~latency of the original inference)
> 4) Sparse completion removal (milliseconds)
> 5) Dimensionality reduction and clustering (depend on the algorithm and compute power, but because of step 4 they are typically done on only tens/few hundreds of data points)
>
> 2+5 are our current bottlenecks, in terms of latency. For our proof-of-concept we did not optimize these steps for latency, and we believe it can be greatly improved. We will add a discussion of factors affecting latency to the manuscript.
>
>
> * Line 385 (“Which parameters are being finetuned?”)
>
> We apologize, fine-tuning was not the correct term here. The parameters tuned are not BERT’s. We meant to write that we perform a hyperparameter search, for setting hyperparameters such as how many completions to retrieve, the function/threshold for sparsity, which scoring functions to use for the clustering, etc. (see Appendix “RQ3: implementation details”, and also in the repository we will release upon acceptance). We will fix this in the manuscript.

---

### Official Review · Reviewer_YpEe · 2023-08-05

**Soundness:** 2

**Excitement:**

3: Ambivalent: It has merits (e.g., it reports state-of-the-art results, the idea is nice), but there are key weaknesses (e.g., it describes incremental work), and it can significantly benefit from another round of revision. However, I won't object to accepting it if my co-reviewers champion it.

**Paper Topic And Main Contributions:**

This paper engages with the interaction between large language models (LLMs) and human concepts. The authors conducted an analysis of contemporary LMs, examining how well these models capture and organize human concepts. They uncovered inconsistencies in these models and identified areas where LMs diverge from human organization of concepts. Moving from analysis to application, they proposed an approach for developing concept-aware LLMs that involves pre-training LLMs with concepts. This represents a shift from the traditional token-based processing to a concept-based approach. They presented a proof-of-concept model which, despite its simplicity, was able to enhance the ranking and robustness of LLMs without requiring additional training. Finally, the authors highlighted potential avenues for future research, suggesting that integrating concept-awareness in LLMs could improve numerous downstream tasks such as learning, planning, and reasoning. This work thus offers a fresh perspective on the intersection of LLMs and human concepts.

**Reasons To Accept:**

- The paper provides an innovative exploration of the interaction between large language models (LLMs) and human concepts, opening up a relatively unexplored area of research.

- The paper proposes an interesting approach for developing concept-aware LLMs, which involves pretraining these models using concepts, a novel shift from token-based to concept-based processing.

- The proof-of-concept model developed in the paper manages to enhance the ranking and robustness of existing LLMs without the need for additional training, suggesting practical applications of the research.

**Reasons To Reject:**

- Despite claiming that large language models (LLMs) do not align well with human concepts, the paper doesn't validate this claim adequately as it doesn't test its hypotheses on state-of-the-art LLMs like ChatGPT, GPT-4, or Llama.

- The paper seems to be initially developed with a focus on language models (LMs), with some cosmetic changes made to tailor it for LLM theme track. Evidence for this includes the naming of the model column as "LM" instead of "LLM" in Table 1. Moreover, the comparative study between BERT and concept-BERT in Tables 2 and 3 doesn't convincingly demonstrate the concept's application in the context of state-of-the-art LLMs. The experimental design, overall, appears more suited for masked language models like BERT, which raises doubts about the paper's alignment with the LLM theme.

- The paper lacks a comprehensive analysis or understanding of the practical challenges and potential privacy concerns of obtaining and using concept-level data for training large language models.

**Reproducibility:**

3: Could reproduce the results with some difficulty. The settings of parameters are underspecified or subjectively determined; the training/evaluation data are not widely available.

**Reviewer Confidence:**

4: Quite sure. I tried to check the important points carefully. It's unlikely, though conceivable, that I missed something that should affect my ratings.

---

> ### Author Rebuttal · Authors · 2023-08-29
>
> Thank you for your comments. We are glad you found our exploration innovative and the proposed approach interesting and novel, with possible practical applications.
>
> * “[T]he paper doesn't validate this claim adequately as it doesn't test its hypotheses on state-of-the-art LLMs like ChatGPT, GPT-4, or Llama.”
>
> There are countless LLMs to try, and we chose several representatives from different model families. Specifically, we used GPT-davinci-003, which is a sibling of ChatGPT (both based on GPT-3.5).
> As for GPT-4, it was made publicly available via OpenAI’s API on June 6th (https://openai.com/blog/gpt-4-api-general-availability) – less than 2.5 weeks before the submission deadline for EMNLP. We would gladly add results for it to the manuscript.
>
> (We note that the recent GPT models do not provide information regarding their training data and architecture, which makes it difficult for the scientific community to evaluate, except as a black box. Having said that, we acknowledge that these models are SOTA, and thus we chose a representative from this family.)
>
>
> * LM vs. LLM, alignment with the theme track:
>
> In line with the spirit of the theme track (and in particular, the question “How do LLMs capture world knowledge?” and the invitation for all kinds of contributions, including position papers), we presented here an approach for developing concept-aware LLMs as well as a proof-of-concept. Our main goal is to raise awareness and spur new research around concept-aware LLMs; this work was _not_ meant to be a comprehensive analysis of all contemporary LLMs.
>
> We chose BERT for our proof-of-concept as it is (still today) the most popular model on HuggingFace for the fundamental fill-mask task. Our pipeline supports any model that can return a ranked list of completions, their embeddings and perplexity scores (and indeed, we have already achieved promising initial results for T5 that we could add to the paper); GPT does not provide the access we need, and thus we have not included it in the comparative study.
>
> We note it is possible to extend our framework to black-box models (e.g., by sampling multiple completions and rethinking the aggregation function), but we leave this for future work.
>
> (In Table 1 that was just a typo. Thanks for pointing it out.)
>
> * “The paper lacks a comprehensive analysis or understanding of the practical challenges and potential privacy concerns of obtaining and using concept-level data for training large language models”
>
> We will gladly expand on practical challenges (we did discuss the main practical challenges for approach 2 in the limitations section). As for privacy concerns, we believe our approach does not add any major concerns on top of those already raised by using LLMs, as we are not using any private data. In particular, approach 1 uses publicly available data such as Wikipedia categories or knowledge graphs. Approach 2 uses the LLM’s embeddings. The only place we can see private information leaking might be the automatic paraphrasing (if the paraphrasing tool was trained on data such as personal emails), but this is completely orthogonal to our algorithm; we will add a note about this to the manuscript.

---

### Official Review · Reviewer_HKKb · 2023-08-05

**Soundness:** 3

**Excitement:**

3: Ambivalent: It has merits (e.g., it reports state-of-the-art results, the idea is nice), but there are key weaknesses (e.g., it describes incremental work), and it can significantly benefit from another round of revision. However, I won't object to accepting it if my co-reviewers champion it.

**Paper Topic And Main Contributions:**

This paper presents a study on how well LLMs capture human concepts and their organization. The authors perform this analysis by focusing on the “type of” relation which pertains to hyponyms and hypernyms. Their results indicate that while LLMs do indeed capture concepts to some extent, they are still far from the human baseline. The authors also propose approaches to enhance LLMs in terms of concepts.

**Reasons To Accept:**

1. The research questions and motivation of the paper is interesting and timely.
2. Some of the results presented in the paper are interesting (for eg the difference in performance of GPT on hypernym vs hyponym).
3. The paper is well-written and well-presented.

**Reasons To Reject:**

1. The motivation and central questions of the paper talks about concepts at a very broad scale but the actual implementation is limited to hyponyms and hypernyms.
2. The approach 1 of rethinking LLMs training does not make a lot of sense to me. It would go against the fundamentals of “language modeling” since such concepts don’t explicitly occur as special tokens in language utterances. It is also only a sketch not backed by concrete empirical evidence so I am skeptical about its usefulness.
3. It is not immediately clear to me what the benefit would be to shift existing LLMs from token-level to concept-level by following the second approach presented in the paper. The authors do not demonstrate how such a “concept-aware” LLM would be better at language understanding or reasoning than normal LLMs.

**Reproducibility:**

4: Could mostly reproduce the results, but there may be some variation because of sample variance or minor variations in their interpretation of the protocol or method.

**Reviewer Confidence:**

3: Pretty sure, but there's a chance I missed something. Although I have a good feel for this area in general, I did not carefully check the paper's details, e.g., the math, experimental design, or novelty.

---

> ### Author Rebuttal · Authors · 2023-08-29
>
> Thank you for your constructive comments. We are glad you find our research questions and motivation interesting and timely, and our experimental results interesting.
>
> * “The motivation and central questions of the paper talks about concepts at a very broad scale but the actual implementation is limited to hyponyms and hypernyms”
>
> Concepts are indeed a very broad and rich topic that can be tackled from many different perspectives. We started exploring this topic at the most fundamental level – the way concepts are hierarchically organized. This aspect has been extensively studied in cognitive science, and thus provides a good starting point for our exploration: we start by assessing the common, agreed-upon organization principles of asymmetry, transitivity, and property inheritance.
>
> * Approach 1, “go against the fundamentals of language modeling since such concepts don’t explicitly occur as special tokens in language utterances”, “a sketch not backed by concrete empirical evidence”:
>
> We wish to clarify that Approach 1 uses regular natural language text, without any special tokens for concepts; the model’s task is to find concepts that will help it complete the masked text.
>
> We are inspired by the Toolformer paper, which was trained to decide which API calls to use and when (given standard text input, without special tokens) and how to best incorporate the results of the calls into token prediction. Toolformer showed improved performance across various downstream tasks compared to GPT-based models. It demonstrated that LLMs can be taught (in a self-supervised manner) when and how to use tools (in our case, concepts) to enhance performance.
>
> Another reason we believe this approach is promising is that many papers have shown empirical evidence that adding intermediate tasks can actually improve performance on the final task (most famously, the chain-of-thought paradigm).
>
> * “It is not immediately clear to me what the benefit would be to shift existing LLMs from token-level to concept-level by following the second approach presented in the paper”
>
> We base our proof-of-concept on the most fundamental language-model task – text completion, and demonstrate the potential of concepts. (As this is the theme track, we focused more on the high-level idea of concept-aware LLMs and less on downstream tasks.)
>
> We acknowledge there are many downstream tasks to explore in follow-up work. Intuitively, one place where we believe this approach will be particularly useful is whenever there is _disambiguity_ (speech recognition, word sense, spell check…). Consider rare words that need to be identified by automatic speech recognition: by working at concept-level, these words could be clustered together with completions that are likely given the rest of the sentence but do not sound similar to the audio, thus increasing the acoustic model’s certainty score.
>
> Thanks again for your comments and questions. We believe they will considerably improve the paper.

---

### Meta-Review · Area_Chair_9ghM · 2023-09-15

**Recommendation:** 3

**Metareview:**

Scores were mixed: 3,2,4 (soundness) and 3,3,4 (excitement). The authors addressed some concerns, leading to increased scores.

Some of the key strengths and weaknesses identified by the reviewers included:

Strengths:

- The research question and motivation were considered timely and interesting (R1, R2, R3)
- Reviewers found some of the results interesting (R1) and the paper well-written (R1, R2)
- paper proposes a new approach for developing concept-aware LLMs by pretraining using concepts (R2)
- includes promising proof-of-concept results (R2)
- behavioral experiments well-designed (R2)

Weaknesses

- despite very general motivation, the paper focuses on hyponyms and hypernyms (R1)
- concerns about how realistic a shift to concept-level LLMs would really be (R1)
- makes claims about inabilities of LLMs without testing on SOTA LLMs (R2) [a similar concern from R3 was addressed in the author response]

I did not find the soundness-related concerns by the second review (YpEe) to be fully convincing. That review lists three reasons to reject: lack of SOTA LLMs, perceived mismatch with the theme track, and lack of discussion of privacy concerns. Only the first concern appears to be potentially a major soundness concern, but the authors pushed back, noting i.a. that the paper includes GPT davinci-003, which can allay some of this concern.

In conclusion, the reviews suggest that the paper is reasonably sound and was perceived as moderately-to-strongly exciting by the reviewers.

---

### Decision · Program_Chairs · 2023-10-07

**Decision:**

Accept-Findings

**Comment:**

Scores were mixed: 3,2,4 (soundness) and 3,3,4 (excitement). The authors addressed some concerns, leading to increased scores.

Some of the key strengths and weaknesses identified by the reviewers included:

Strengths:

- The research question and motivation were considered timely and interesting (R1, R2, R3)
- Reviewers found some of the results interesting (R1) and the paper well-written (R1, R2)
- paper proposes a new approach for developing concept-aware LLMs by pretraining using concepts (R2)
- includes promising proof-of-concept results (R2)
- behavioral experiments well-designed (R2)

Weaknesses

- despite very general motivation, the paper focuses on hyponyms and hypernyms (R1)
- concerns about how realistic a shift to concept-level LLMs would really be (R1)
- makes claims about inabilities of LLMs without testing on SOTA LLMs (R2) [a similar concern from R3 was addressed in the author response]

I did not find the soundness-related concerns by the second review (YpEe) to be fully convincing. That review lists three reasons to reject: lack of SOTA LLMs, perceived mismatch with the theme track, and lack of discussion of privacy concerns. Only the first concern appears to be potentially a major soundness concern, but the authors pushed back, noting i.a. that the paper includes GPT davinci-003, which can allay some of this concern.

In conclusion, the reviews suggest that the paper is reasonably sound and was perceived as moderately-to-strongly exciting by the reviewers.